# An Integrated Decision Support System for the Sustainable Reuse of the Former Monastery of "Ritiro del Carmine" in Campania Region

**Francesca Torrieri** [1,*] , **Marina Fumo** [2] **, Michele Sarnataro** [2] **and Gigliola Ausiello** [2]

1    Department of Industrial Engineering, University of Naples Federico II, Piazzale Tecchio 80,
     80125 Naples, Italy
2    Department of Civil, Architectural and Environmental Engeneering, Piazzale Tecchio 80, 80125 Naples, Italy;
     marina.fumo@unina.it (M.F.); michele.sarnataro@gmail.com (M.S.); ausiello@unina.it (G.A.)
*    Correspondence: frtorrie@unina.it

**Abstract:** Nowadays, many public administrations have abandoned and underused heritage buildings due to a lack of public resources, although the effective contribution of cultural heritage as a driver and enabler of sustainable development is strongly recognized. Currently, investments in cultural heritage have multidimensional impacts (social, economic, historical, and cultural) and can contribute to increasing overall local productivity; improving the wellbeing of inhabitants; and attracting funding from the public, private, and private–social sectors. Lack of public resources has pushed local administrations to favor new forms of valorization of public property that can promote the "adaptive reuse" of historic buildings in order to preserve their social, historical, and cultural values. At the same time, administrations seek to stimulate the experimentation of new circular business, financing, and governance models in heritage conservation, creating synergies between multiple actors; reducing the use of resources; and regenerating values, knowledge, and capital. The objective of this paper is to propose an integrated evaluation model, based on multicriteria analysis, and a financial model to support the choice of an alternative reuse of an ancient monastery in the municipality of Mugnano in the Campania region in order to define a "shared strategy" based on a "bottom-up" approach. This starts from the needs of the local community but does not neglect the historical and cultural values of the heritage building, as well as the economic and financial feasibility. The positive results obtained show that the model proposed can be a useful decision support tool in environments characterized by high complexity such as cultural heritage sites, where the objective is to precisely highlight the elements that influence the dynamics of choice for building shared bottom-up development strategies.

**Keywords:** integrated evaluation; multicriteria analysis; cultural heritage and circular economy; financial sustainability; Ritiro del Carmine; built heritage sustainable reuse

## 1. Introduction: The Sustainable Reuse of Cultural Heritage from a Circular Economic Perspective

In the current European context, the conservation of cultural heritage presents a real challenge for professional and public institutions at national and local levels, even if it is strongly recognized that the heritage sector makes a significant economic contribution [1–3].

In Italy, many public institutions have an enormous asset that is composed of several unused buildings [4]. In recent years, difficulties in finding new uses for these properties has led to their abandonment and ruin.

Lately, Italian authorities have experimented with new approaches to valorize these massive estates, which are otherwise destined to remain unused [5]. The new policies look at redevelopment as

a possibility to enhance public welfare through the creation of new social hubs [6] and a new sense of community within the population [7]. The promotion of these types of actions, strictly linked to citizens' social needs and social capital [8,9], allows the development of more sustainable and successful strategies [10]. In fact, investments in cultural heritage have multidimensional impacts (social, economic, historical, and cultural) and can contribute to increasing overall local productivity; improving inhabitants' wellbeing; and attracting funds from the public, private, and private–social sectors [11,12]. As Throsby highlights, cultural heritage can be considered as the "glue" between the different dimensions of sustainable development [13].

The redevelopment of ancient and unused buildings represents an opportunity to pursue new and innovative solutions [14].

In this context, the concept of adaptive reuse plays a significant role. Adaptive reuse helps to protect and preserve historical buildings against obsolescence, considering environmental, social, and financial aspects of sustainability and promoting the valorization of the surrounding society [15–17].

Currently, the conservation of cultural heritage can promote social cohesion and integration through regeneration of neglected areas, creation of locally rooted jobs, and promotion of a shared understanding and sense of community [18]. In this sense, cultural heritage must gain an active role both in today's society and the urban reality, especially in many small town and historic centers, as the case study presented here well represents.

Moreover, from a circular economic perspective, the efficient reuse of cultural heritage and the choice of recycled and natural materials help to reduce negative externalities and produce positive environmental, social, and cultural impacts which benefit the whole society [19].

The choice of functions compatible with the building structure and historical values, expressing the needs of local communities, helps to integrate cultural assets within the city context and to attract financial capital to ensure its management over time. Moreover, the use of recycled and natural materials is the starting point of a circular path that follows the environmental, economic, and social sustainability perspective [20–22].

The choice of a new function requires a systematic framework to evaluate the different feasible alternatives and sufficient information to identify the best solution, or at least, the best compromise solution [23]. The evaluation process has to handle the problem holistically by considering different perspectives, objectives, stakeholders, and values in a comprehensive manner; such a process may increase the quality of public decisions [24].

In this context, the objective of the paper is to propose an integrated evaluation model, based on multicriteria analysis, and a financial model to support the choice of the alternative reuse function of an ancient monastery in the municipality of Mugnano in the Campania region in order to define a "shared strategy" based on a bottom-up approach. This starts from the needs of the local community but does not neglect the historical and cultural values of the heritage building, as well as the economic and financial feasibility.

The methodology follows the general approach to decision problems [25], adapted to the case study analyzed in order to support the public administration—the owner of the monastery—to choose the best alternative reuse functions from a sustainable perspective.

The paper is organized as follows. In Section 2, we illustrate the framework adopted. In Section 3, we report the results of the social and financial evaluations tested on the case study of the Ex Ritiro del Carmine. In Section 4, the conclusion and discussion of future research are presented.

## 2. Material and Methods

### 2.1. The Integrated Decision Support System for the Choice of Alternative Functions

The redevelopment of heritage buildings is a complex design problem, in which several points of view need to be managed in a holistic evaluation process.

Following the classical approach proposed by Simon (1972) to the decision-making process, an integrated methodological framework was developed for this case study, as Figure 1 illustrates.

Simon defines three main stages of the decision-making process:

- Intelligence, which deals with the problem of identification of and data collection for the problem.
- Design, which deals with the generation of alternative solutions to the problem at hand.
- Choice, that is, selecting the "best" solution from among the alternative solutions using some criterion.

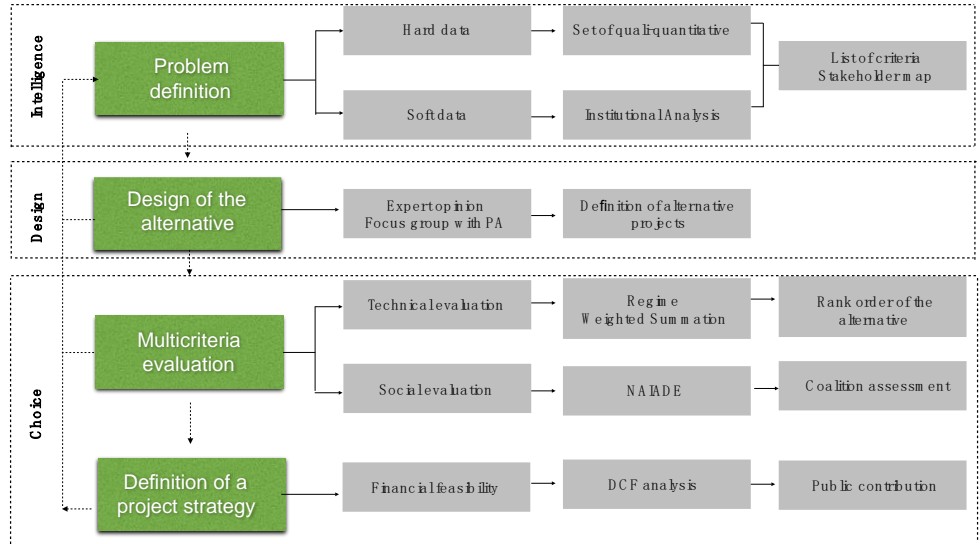

**Figure 1.** Methodological framework.

Starting from the main stages proposed by Simon, different evaluation methods have been proposed in order to support the definition of different reuse alternatives in redevelopment projects, including both technical problems related to the design projects [21–23,25], social issues, and financial feasibility. In the stated methodological framework, different multicriteria methods are combined with a financial model in the diverse stages of the decision-making process in order to develop a tool to enhance the quality and trustworthiness of the decision-making process itself.

The methodological framework is illustrated in Figure 1.

In the first stage of the decision-making process, the decision maker (DM) identifies problems, opportunities, and objectives regarding the reuse project of the heritage building. To accomplish this, a list of criteria was identified. The list of criteria was structured based on appropriate reference [26], which identified the main categories of analysis, but considering the peculiarity of the case study, this also emerged in a focus group with the public administration and the technicians.

Moreover, a map of relevant stakeholders was defined in accordance with the strategic objectives of the public administration oriented toward the valorization of the building as having symbolic value and being a meeting place for the local community.

Then, in the design phase, five alternatives were identified during a focus group organized by the municipality with the experts (architect, economist, and sociologist) and politicians.

The alternative solution was evaluated by means of a multicriteria method considering two types of analysis, namely a "technical evaluation" linked with the DM's preferences and a "social evaluation" linked with the stakeholders' points of view [27,28].

The technical evaluation was developed with the aim of defining a rank order of alternatives, considering the list of criteria previously defined with the help of two multicriteria methods, namely, regime analysis [29] and the weighted summation method (WSM) [30].

The regime method is used to start a dialogue between the DM and the experts, which has the advantage of simplifying the debate with the DM: at the beginning, he often does not have a clear idea

of the problem and is not familiar with the decision-making tools. The regime method is a discrete multicriteria method with a partial compensatory structure based on pairwise operations. This method can handle mixed types of information, that is, both quantitative and qualitative. The fundamental input data are the impact matrix and the political weights, and these two elements are combined to calculate the probability that one alternative is preferred over another. The impact matrix includes the evaluation of each alternative with respect to each criterion. The set of weights is a qualitative assessment consisting of an ordinal evaluation of the criteria reflecting the DM's preferences.

Weighted summation is then used to check the previous results through the improvement of the set of weights: it regards a more complex process based on the pairwise comparison of the decision criteria. In this part, the main information is provided for better comprehension of the problem, the opportunities, and the potential of the alternative projects. Weighted summation is a compensatory approach based on a linear model. This method uses an indirect approach, in which the qualitative information is first transformed into cardinal information to be compared; this procedure is called "normalization" and it is necessary to handle different types of attributes. The input data are the impact matrix and the matrix of weights. The evaluations of each alternative with respect to each criterion are combined through the matrix of weights in one overall value, obtained by the addition of all the weighted scores together. The combination of the two previously described methodologies encourages a transparent dialogue between the public administration, which owns the building, and the architects in order to define a sustainable solution taking into account the different dimensions involved. Moreover, the methodologies proposed are included in the Definite software, which makes communication easier among different actors.

The Definite program was developed by Ron Janssen and Marine van Herwijnen in 1987 [31]. It is a multiobjective decision support system that supports the whole decision process from problem definition to report generation. The system performs the following functions: (1) structure the problem and generate alternatives; (2) compare alternatives; (3) rank and/or value alternatives; (4) support interpretation of the results; (5) present results. To perform these functions, the system contains five modules. Each module contains a variety of procedures (like the regime and weight summation methods) to perform these functions.

The social evaluation was developed with the aim of understanding the possible coalition generated by the choice of the best alternative [32,33]. This analysis was carried out with the help of the Novel Approach to Imprecise Assessment and Decision Environments (NAIADE) method [34]. The NAIADE method captures the preference of stakeholders and supplies indications of the distance of positions among the different interest groups. It evaluates the social compromise solution through the analysis of the possible coalitions. The NAIADE method is a discrete social multicriteria method, which includes mixed types of information and a conflict analysis in a fuzzy environment. Through pairwise linguistic evaluation, "based on semantic distance between linguistic qualities", two types of evaluation are provided. The first regards the assessment of the alternatives based on the social impact matrix, which contains a qualitative evaluation of each alternative with respect to a defined set of criteria, based on the stakeholder's preferences. The second analysis is performed by the completion of an equity matrix, where a similarity matrix is calculated. It sheds light upon the level of decision conflicts among the different interest groups and highlights the possible formation of coalitions (building a dendrogram of coalitions), showing the impact of each alternative as perceived by the social actors. In this way, NAIADE provides the following information: (a) distance indicators between the interests of the different social actor groups, as an indication of coalition formation possibility or interest convergence; (b) rankings of alternatives for every coalition, in accordance with the impacts on the social groups or the social compromise solution.

Then, the preferred alternative is developed, and the financial feasibility is evaluated. The method combines different assessment phases with sequential checks. This kind of structure allows the creation of an interactive framework able to incorporate the ideas of learning processes and the engagement of

different stakeholders. These characteristics may provide better-informed decisions and a greater level of consent.

Multicriteria decision analysis (MCDA) can handle the complexity of the whole process. The MCDA methods provide tools for gauging stakeholders' preferences, comparing alternatives, and supplying useful indications to the DM [35]. From this perspective, MCDA has the principal aim to "create" instead of "find" solutions; therefore, it is a "constructive" approach [36].

Multicriteria analysis has the ability to compare the alternatives according to various conflicting stakeholder interests. The ability to involve several points of view in the early stage of the design problem through a participative process may help to avoid conflict and make more successful and transparent decisions [37,38].

Actually, the evaluation process is not a "one-shot activity"; instead, it is a constructive, dynamic process that advances in relation to continuous reinforcing gains along the various steps [39–43].

## 2.2. Introduction to the Case Study

The case study under analysis regards the reuse of the unused monastery Ex Ritiro del Carmine in the Campania region (Italy).

The ancient building is located in the old town of the municipality of Mugnano di Napoli, a town located in the northern area of the Metropolitan City of Naples. The city has approximately 35,000 inhabitants and is directly dependent on the City of Naples, where the main activities and public services are located. The municipality's future development strategy establishes the conservation and enhancement of the old town together with the creation of new residential areas and public services [44].

The monastery was built as an orphanage for girls in 1818 [45,46], far from the city center of that time. The growth of the congregation led to incremental construction: In 1860, the ground floor was completed, and the Church of Santa Maria del Carmine was built as an annex to it. In 1937, the original order left the building, which was acquired by a new congregation who enlarged the monastery building with two other floors and established a private elementary school. The order left the monastery in 2003 after many years of intense activity. The municipality acquired the monastery in 2010, even though it has been unused since 2003.

As mentioned above, the building is located in the old town, far from the main roads, in a quiet place alongside a by-road. The complex is composed of four main parts: the monastery, the church, the theater, and the garden. It is important to underline that the church is still open and is not part of the municipality project (Figure 2).

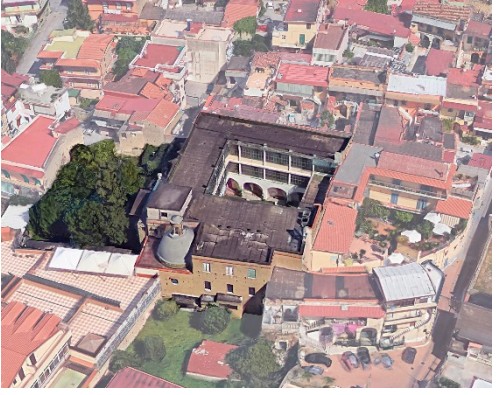 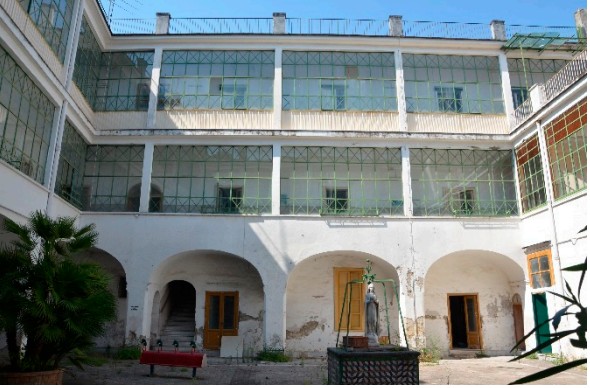

**Figure 2.** The monastery of Ex Ritiro del Carmine in the Campania region (Italy).

The monastery itself has three levels and a central courtyard, while the garden is on the western side together with the theater. On the ground floor (approx. GEA 1000 m$^2$), there are the entrance hall, the reception, the porch, the kindergarten, the kitchen, and the canteen; on the first floor (approx.

GEA 850 m$^2$), there are the nuns' bedrooms; on the second floor (approx. GEA 800 m$^2$), there are the classrooms and two other bedrooms.

## 3. Results

### 3.1. Problem Definition

The definition of the criteria is a process, which follows a hierarchical logic. Starting from the dialogue with the DM, the list of subcriteria are formulated by the technicians and reflect the DM's objectives and needs. For each fixed criterion, more specific elements are defined: the subcriteria are measurable and index-linked with the specific dimensions of the projects.

In our case study, the DM showed the intention of pursuing the social-sustainability-based criteria mentioned by the most appropriate references [13,47,48]. This vision is elaborated in a holistic view and includes the enhancement of social objectives, economic objectives, and urban development. At this high-level stage of project development, no specific environmental issues are considered. Environmental implications should be addressed at the more refined second stage of analysis.

The hierarchical structure of the criteria was formulated by a focus group, in which the different points of view arose and became clearer. The board session allowed the technicians to summarize the information in a limited number of clearly defined criteria. For the analyzed case study, the formulated criteria are represented in a tree chart, as shown in Figure 3.

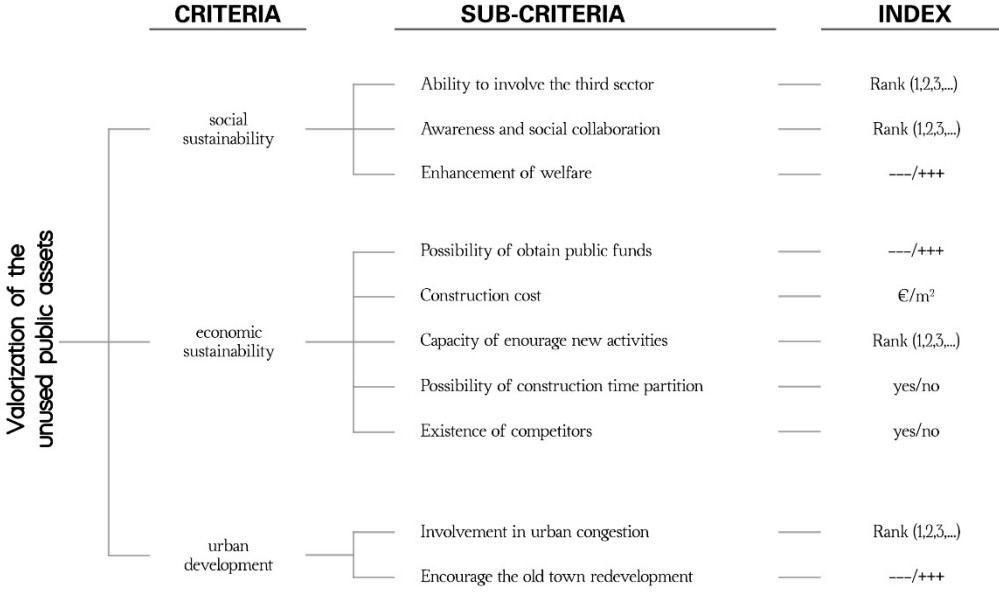

**Figure 3.** Tree chart of criteria.

As Figure 3 illustrates, the criteria identified are more oriented toward social issues and the integration of the building in the urban context. Currently, the main strategic objective of the public administration is to create a new catalyst for the entire urban development process.

The evaluation subcriteria are described below:

- Ability to involve the third sector: assesses the availability of nonprofit organizations to take part in such activities;
- Awareness and social collaboration: linked with the capability to involve citizens in social activities and attract interest in social issues;
- Enhancement of welfare: assesses the ability to answer social needs;
- Possibility of obtaining public funds: linked to the availability of public grants;

- Construction cost: a parametric appraisal of the possible square-meter cost in relation to the specific characteristics as defined in the regulation;
- Capacity to encourage new activities: linked with private investment in new activities not necessarily directly linked with the project;
- Possibility of construction time partition: assesses the possibility of splitting the construction into several periods;
- Existence of competitors: assesses the existence of other similar activities in the municipality;
- Involvement in urban congestion: linked with the growth of traffic overcrowding;
- Encourage old town redevelopment: linked with the possibility of attracting new public and private investment for the redevelopment of the old town.

In a public project, key stakeholders have to be involved in order to help the DM in making better decisions and improving organization performance. The redevelopment of the monastery aims to enhance the welfare of the whole population of the municipality. For this reason, the population is divided into several groups in order to involve every social party.

As defined in [49], stakeholders are any group of people, organized or not, who share common interests, values, and behaviors and who can affect or be affected by the outcomes of the project. The representative groups are selected by the technicians to address all the social components involved in the project [50].

Here, the selected groups were

- public administration;
- political opposition;
- entrepreneurs;
- freelance professionals;
- business owners;
- social and cultural associations;
- students;
- the employed;
- the unemployed;
- the retired.

These groups were involved in the process to evaluate the developed alternative in order to understand the level of consensus among them on the rank order evaluation.

### 3.2. Design of the Alternative

In the early stage of the evaluation process, with respect to the objectives of the public administration, the designers formulated three alternatives. These are briefly described below.

**Alternative A: Antiviolence Center for Women**

This alternative is in response to the increasing phenomenon of violence against women observed by the municipal "antiviolence desk" and by the Italian National Institute of Statistics (ISTAT) [51]. The project would provide hospitality for both women and their underage children who were victims of or were exposed to violence. The center would grant several services, including expert consultation, legal advice, and psychological assistance and support groups. It would organize information meetings and public events to spread knowledge and tackle the spread of the problem.

**Alternative B: Refugee Center**

This alternative is in response to the "European migrant crisis" characterized by the increase in migrants arriving in the European Union. The project aims to provide a place in which the refugees may be hosted, and it would grant hospitality, social services, and legal support. The center may

organize events and public meetings to promote the idea of a new multicultural society based on equality between people.

**Alternative C: Cultural Center and Library**

This alternative is in response to the lack of public social and cultural spaces, especially a public library in the municipality. The project aims to create a center of excellence to support citizens' education that would gather public services, study rooms, laboratories for shared workshops, and a conference room.

The redevelopment of the theater and the garden would provide two new recreational areas for entertainment and outdoor activities.

The above-presented alternatives derive from a static vision in which a single function absorbs the whole complex. These three alternatives were presented in a focus group to the public administration and the new idea emerged in a brainstorming section considering the characteristic of the building and the objectives of the public administration.

Actually, during the deliberation process, the possibility of taking advantage of the characteristics of the complex and a multifunctional building was suggested: the integration of different functions and resources in a hybrid building can enhance the social services provided and reduce public expenditure.

In particular, the Ex Ritiro del Carmine has different characteristics, which support this idea:

- the partition in three levels allows for there to be three different activities with different modes and periods of operation. Each one would be separated from the others, but they would share the courtyard and the garden as places of social exchange;
- the courtyard represents a cornerstone which links all the activities and stakeholders together, creating a place of meetings and cultural exchanges;
- the shape of each story gives some suggestions about the use of the level: the ground floor could host shared functions with a food service; the first floor, which included the nuns' room, could host hospitality functions; the second floor, which has big free rooms, could host the library.

These features characterize the new formulated alternatives that present the integration of different activities, in particular, the organization rooms on the ground floor, the hospitality activities on the first floor, and the library on the second floor. The two other proposed alternatives differ in terms of the hospitality activities. They are presented below.

**Alternative D: Social Hub with Refugee Center**

This project aims to create a center in which to develop new procedures of cultural integration between refugees and citizens, which would help refugees and tackle the widespread forms of racism.

**Alternative E: Social Hub with Antiviolence Center for Women**

This project aims to create a multifunctional center in which the female guests would recover their independence through new forms of work integration. The center would also organize seminars to spread knowledge about the phenomenon of violence.

The five developed alternatives were then evaluated in a second stage of the decision process. They were evaluated from a technical point of view considering the list of criteria developed and with the help of regime analysis and weighted summation. Then, the social acceptability of the solution found was evaluated with the NAIADE method. The results obtained are reported in the next subsection.

*3.3. Multicriteria Analysis*

3.3.1. Regime Method

The regime method is based on two input data: the impact matrix and the weight vector to compare the alternative and to define a ranking of these.

The impact matrix developed for the analyzed case study is illustrated in Table 1, starting from the list of criteria identified in Figure 1.

**Table 1.** Impact matrix.

| | | Subcriteria | Index | c/b | Project Alternatives | | | | |
| | | | | | A | B | C | D | E |
|---|---|---|---|---|---|---|---|---|---|
| CRITERIA | social | Ability to involve the third sector | rank | b | 2 | 5 | 3 | 4 | 1 |
| | | Awareness and social collaboration | rank | b | 1 | 2 | 5 | 4 | 3 |
| | | Enhancement of welfare | —/+++ | b | ++ | + | ++ | ++ | +++ |
| | economic | Possibility of obtaining public funds | —/+++ | b | ++ | 0 | 0 | + | ++ |
| | | Construction cost | €/m$^2$ | c | 793 | 793 | 975 | 904 | 904 |
| | | Capacity of encouraging new activities | rank | b | 5 | 4 | 1 | 3 | 2 |
| | | Possibility of construction time partition | yes/no | b | no | no | yes | yes | yes |
| | | Existence of competitors | yes/no | c | no | yes | no | yes | no |
| | urban | Involvement in urban congestion | rank | b | 3 | 1 | 5 | 2 | 4 |
| | | Encourage the old town redevelopment | —/+++ | b | — | – | ++ | + | ++ |

As illustrated in Table 1, in order to evaluate the alternatives and to reflect the complexity of the design problem, each criterion uses a different scale of measurement, qualitative or quantitative, due to their nature and the available information. It is important to highlight that each criterion can be considered as a cost or a benefit, and for this reason, they have to be respectively minimized or maximized.

The main scales are nominal, ordinal, binary, and ratio. For simplicity, we refer to the first three as qualitative information and to the last one as a quantitative scale.

In the case of the Ex Ritiro del Carmine, the scale of measurement and the related unit of measurement used are

- rank: ordinal scale, ranks the alternatives with respect to the analyzed criterion; it goes from 1, for the best alternative, to 5, for the worst one;
- —/+++: nominal scale, consists of a linguistic evaluation with respect to a seven-level qualitative scale; the levels are high (+++/—), medium (++/–), and low (+/-), for both positive and negative values, and "0" for moderate;
- €/m2: ratio scale, a statistical appraisal of the unitary cost of construction for each alternative;
- Yes/no: binary scale, it defines the existence of a certain condition for each evaluated alternative.

For each criterion, it is indicated whether the criterion represents a cost or a benefit (c/b).

The system of weights identifies the priority among the evaluation criteria, which is basically a political issue linked to the policy game and anyone involved in the decision-making process.

The weight vector consists of an ordinal assessment of the criteria. This type of evaluation allows for the simplification of the dialogue with the DMs, especially in the early stage.

During the decision-making process, the DM was asked to express his own preferences through the assessment of the set of weights associated with the evaluation criteria. The identified ordinal sets of weights are shown below in Figure 4.

The two sets highlight two different approaches: the first, which refers to the political class, attributes more importance to the social aspects, while the second, which refers to the technical experts, pays close attention to the economic feasibility.

The results obtained with the regime method are shown in Figure 5.

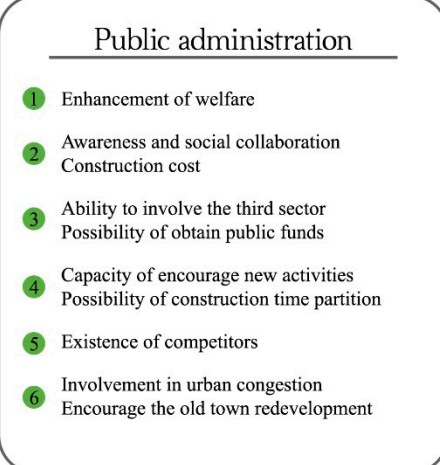

**Figure 4.** Rank order of criteria.

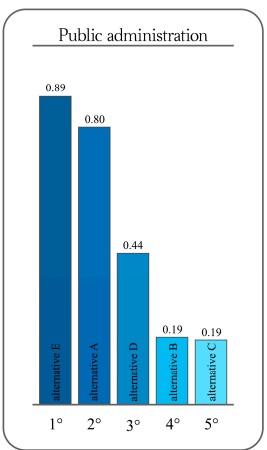
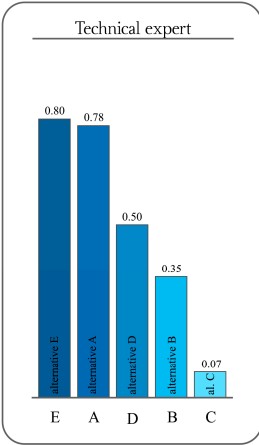

**Figure 5.** Results of the regime method: rank order of alternatives.

Both the rankings presented "Alternative E: Social hub with antiviolence center for women" as the best compromise solution. The other alternatives were in the same order, but they presented different scores.

### 3.3.2. Weighted Summation Method

The sensitivity of the rank order obtained with the regime analysis was then tested using the WSM, evaluating two new sets of weights through a pairwise process and using them to rank the alternatives.

The WSM is based on the impact matrix and a set of weights to compare the alternatives and to define a ranking of these. The set of weights consists of a matrix, in which the quantitative evaluation of the DM's preferences is reported; this type of evaluation allows the improvement of the design of the expressed preference through a quantitative pairwise evaluation process shown below (Table 2).

The application of weighted summation allows for verification of the robustness and stability of the results through a "sensitivity analysis". This process assesses the variation of the previous results through the variation of the weight matrix.

The weight matrix is assessed in a process of pairwise comparison among every possible couple of criteria. The DM is asked to express his preference through a numeric value that goes from 1, when the considered criteria have the same importance, to 9, when a criterion is "extremely more important" than the other one. The matrix is not symmetrical, because if we compare *a* with *b*, then the value of *b*

with respect to *a* represents the reciprocal. For example, in our case, the criterion *a* was more important than *b*, and the value was 1.2. So, the value of *b* with respect to *a* was the reciprocal (1/1.2, or 0.833).

The identified weight matrixes are shown below in Table 2.

**Table 2.** Sets of weight matrix.

| Public administration | | a | b | c | d | e | f | g | h | i | l |
|---|---|---|---|---|---|---|---|---|---|---|---|
| Ability to involve the third sector | a | | 1.200 | 0.588 | 1.000 | 0.714 | 1.200 | 1.000 | 1.500 | 1.500 | 1.700 |
| Awareness and social collaboration | b | 0.833 | | 0.500 | 1.300 | 1.000 | 1.400 | 1.700 | 2.000 | 1.800 | 2.200 |
| Enhancement of welfare | c | 1.700 | 2.000 | | 3.000 | 1.500 | 2.000 | 2.000 | 3.000 | 3.300 | 4.000 |
| Possibility of obtain public funds | d | 1.000 | 0.769 | 0.333 | | 0.588 | 1.500 | 1.600 | 2.000 | 2.000 | 3.000 |
| Construction cost | e | 1.400 | 1.000 | 0.667 | 1.700 | | 2.000 | 1.600 | 2.000 | 2.100 | 3.000 |
| Capacty of encourge new activities | f | 0.833 | 0.714 | 0.500 | 0.670 | 0.500 | | 1.000 | 1.500 | 1.600 | 1.800 |
| Possibility of construction time partition | g | 1.000 | 0.588 | 0.500 | 0.625 | 0.625 | 1.000 | | 1.400 | 1.800 | 2.100 |
| Existence of competitors | h | 0.667 | 0.500 | 0.333 | 0.500 | 0.500 | 0.667 | 0.714 | | 1.500 | 1.300 |
| Involvement in urban congestion | i | 0.667 | 0.556 | 0.303 | 0.500 | 0.476 | 0.625 | 0.556 | 0.667 | | 1.500 |
| Encourage the old town redevelopment | l | 0.588 | 0.455 | 0.250 | 0.333 | 0.333 | 0.556 | 0.476 | 0.769 | 0.667 | |

| Technical expert | | a | b | c | d | e | f | g | h | i | l |
|---|---|---|---|---|---|---|---|---|---|---|---|
| Ability to involve the third sector | a | | 1.000 | 3.500 | 4.000 | 0.333 | 1.500 | 0.500 | 2.000 | 1.500 | 1.800 |
| Awareness and social collaboration | b | 1.000 | | 0.500 | 2.500 | 0.667 | 1.500 | 0.714 | 0.714 | 2.000 | 1.000 |
| Enhancement of welfare | c | 0.286 | 2.000 | | 1.000 | 1.000 | 2.000 | 1.700 | 1.500 | 3.000 | 1.700 |
| Possibility of obtain public funds | d | 0.250 | 0.400 | 1.000 | | 1.500 | 3.000 | 2.000 | 2.000 | 3.500 | 3.500 |
| Construction cost | e | 3.000 | 1.500 | 1.000 | 0.667 | | 2.000 | 1.500 | 1.700 | 3.000 | 2.100 |
| Capacty of encourge new activities | f | 0.667 | 0.667 | 0.500 | 0.333 | 0.500 | | 0.667 | 0.500 | 1.000 | 0.769 |
| Possibility of construction time partition | g | 2.000 | 1.400 | 0.588 | 0.500 | 0.667 | 1.500 | | 1.000 | 1.700 | 1.500 |
| Existence of competitors | h | 0.500 | 1.400 | 0.667 | 0.500 | 0.588 | 2.000 | 1.000 | | 2.000 | 1.500 |
| Involvement in urban congestion | i | 0.667 | 0.500 | 0.333 | 0.286 | 0.333 | 1.000 | 0.588 | 0.500 | | 0.714 |
| Encourage the old town redevelopment | l | 0.556 | 1.000 | 0.588 | 0.286 | 0.476 | 1.300 | 0.667 | 0.667 | 1.400 | |

The results of the calculation for the above sets of weights are shown in Figure 6. Both the evaluation rankings presented "Alternative E: Social hub with antiviolence center for women" as the best compromise solution, while the other positions were different in each ranking. These two rankings were different from the previous results due to an improved assessment of the sets of weights.

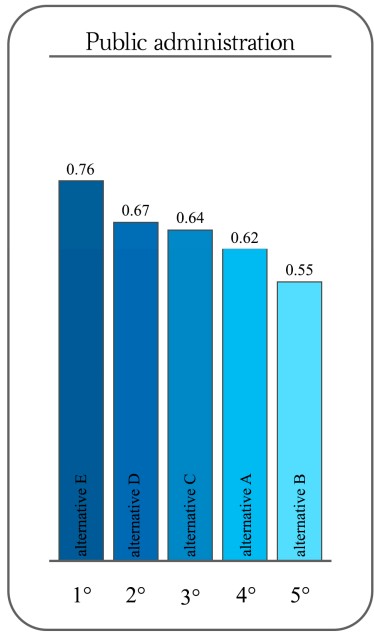 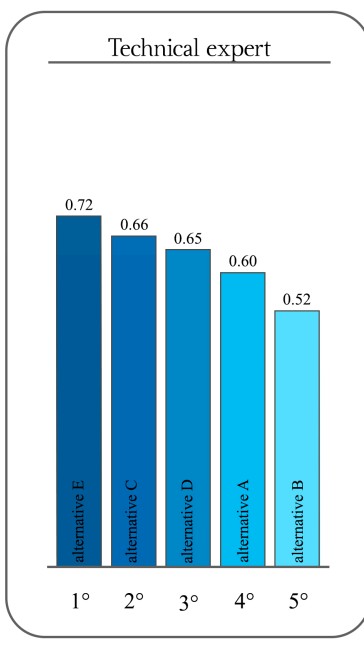

**Figure 6.** Results of the weighted summation method (WSM): rank order of the alternatives.

The results obtained confirmed that the rank order of the alternative is not sensible to the weight vector assessed.

### 3.3.3. Social Evaluation and Coalition Assessment

Starting from the definition of the main social actors (see Section 3.1), the first step was the identification of their points of view in relation to the project alternatives. The information from each group, identified in the first stage of the process, was collected through an online survey supplied through Google Forms. The questionnaire was structured with close-ended questions; they had a limited number of answers, among which the respondent must choose the one which best matches his/her opinion. The form presented an initial introductory part with the aim of obtaining general information about the respondent; the second part described the case study and presented the project alternatives in simple and familiar language; in the third section, the respondent was asked to evaluate the presented alternatives through a linguistic evaluation. The linguistic assessment was expressed on a nine-level qualitative scale from "perfect" to "extremely bad".

Examples of the questions posed in each part of the questionnaire are here reported.

1) Are you a citizen of Mugnano di Napoli?

   a) Yes, I am
   b) I usually frequent the city
   c) No, I am not

2) Do you know the ex-monastery of the Ritiro del Carmine

   a) Yes, I do
   b) I have heard of it
   c) No, I don't

3) How do you evaluate the scenario "C: Cultural center and library" from 1 (perfect) to 9 (extremely bad)?

The collected evaluations were gathered and are displayed in the equity matrix shown in Table 3.

**Table 3.** Equity matrix.

| Stakeholders | Project Alternatives | | | | |
|---|---|---|---|---|---|
| | **A** | **B** | **C** | **D** | **E** |
| Public administration | good | more or less good | good | more or less good | very good |
| Political opposition | moderate | moderate | very good | very good | more or less good |
| Entrepreneur | more or less bad | moderate | moderate | very good | perfect |
| Freelance professional | good | moderate | very good | more or less good | very good |
| Business owner | moderate | moderate | perfect | more or less good | good |
| Social and cultural association | good | more or less good | good | more or less good | very good |
| Student | more or less good | moderate | good | more or less good | very good |
| Employed | moderate | moderate | good | more or less good | very good |
| Unemployed | very good | moderate | good | moderate | good |
| Retired | more or less bad | moderate | very good | more or less good | very good |

The equity matrix provided the linguist indication of the interest group judgments for each alternative. Semantic distance was also used in this case to calculate the similarity indexes among interest groups. A similarity matrix was then computed starting from the equity matrix. The similarity matrix provided an index, for each pair of interest groups i,j, of the similarity of judgement over the

proposed alternatives. This index sij was calculated as sij = 1/(1 + dij), where dij is the Minkovsky distance between groups i and j, which was calculated as follows:

$$dij = \sqrt[P]{\sum_{K=1}^{N} (S_k\ (i,j))^P}$$

where *S(i,j)* is the semantic distance between groups i and j in the judgement of alternative *k, N* is the number of alternatives, and *p* > 0 is the parameter of the Minkovsky distance. Lastly, through a sequence of mathematical reductions, the dendrogram of coalition formation was built. It shows possible coalition formation for decreasing values of the similarity index and the degree of conflict among interest groups.

For the case study, we used the NAIADE software for the calculations. This software allowed for comparison of the preferences expressed and analysis of the similarity between the interests of each group. Furthermore, it supplied a graphic representation of the potential coalitions between the parties in the form of a dendrogram. Each meeting point has an associated numeric value, named "similarity index", which shows the similarity of the coalition, with values between 0 and 1.

In the case of the Ex Ritiro del Carmine, the dendrogram (Figure 7) highlights good results in terms of agreement among the stakeholders.

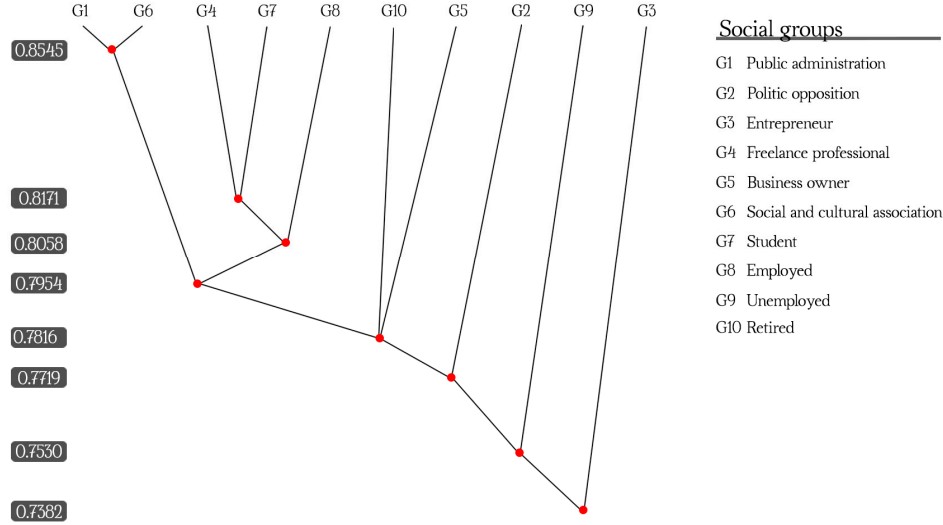

**Figure 7.** Dendrogram of the coalitions.

The first coalition, between G1 and G6 (public administration and social and cultural association), had a similarity index value of 0.8545. The second coalition, between G4 and G7 (freelance professionals and students), had a the similarity index value of 0.8171. The third coalition, between G8 (the employed) and the previous coalition (G4–G7), had a value of 0.8058. The coalition between the two abovementioned groups (G1–G6 and G4–G7–G8) had a value of 0.7954.

In Table 4, the possible coalitions referring to different rankings of the alternatives are shown. It is possible to note that for the ranking (E, C, D, A, B), all the groups were in agreement, having a similarity index of 0.7382, which is really high considering that the maximum value of similarity (total agreement) is 1.

**Table 4.** Possible coalitions on the ranking of the alternatives.

|  |  | 0.8545 | 0.8171 | 0.8058 | 0.7816 | 0.7716 | 0.7530 | 0.7382 |
|---|---|---|---|---|---|---|---|---|
| **RANKING** | 1 | E | E | E | E | E | E | E |
|  | 2 | C | C | C | C | C | C | C |
|  | 3 | A | A | D | D | D | D | D |
|  | 4 | D | D | A | A | A | A | A |
|  | 5 | B | B | B | B | B | B | B |

So, the results of the calculation underline that the favorite solution is the "Alternative E: Social hub with antiviolence center for women".

### 3.3.4. Results Comparisons

In the previous sections, we established the rankings for each phase of the proposed framework with different methods. The obtained results supplied the same preference in every stage of the process (Table 5), so that the best compromise solution may be "Alternative E: Social Hub with antiviolence center for women".

**Table 5.** Comparison between different rankings.

| Ranking | Regime Method | | Weighted Summation | | NAIADE |
|---|---|---|---|---|---|
|  | **P.A.** | **T.E.** | **P.A.** | **T.E.** |  |
| 1 | E | E | E | E | E |
| 2 | A | A | D | C | C |
| 3 | D | D | C | D | D |
| 4 | B | B | A | A | A |
| 5 | C | C | B | B | B |

P.A.—public administration; T.E.—technical experts; NAIADE—Novel Approach to Imprecise Assessment and Decision Environments.

In the next subsection, we evaluate the financial sustainability of the alternative chosen from the stakeholders involved in the decision-making process.

### 3.4. Definition of a Project Strategy

Nowadays, the lack of public resources complicates the financing of the redevelopment process. Therefore, public administrations have to assess the public expenditure in order to guarantee economic sustainability throughout the life of a project. Financial analysis allows the evaluation of both the construction cost and the cash flow and calculates the necessary public economic resources to effectively run a reused building [52,53].

Once the new function is defined, it is possible to evaluate the construction cost of the project. This is composed of several rates [50], in particular, technical construction cost (TCC), taxes (corresponding to 21% of TCC), professional costs, preliminary studies and surveys, tender notice, and accidents (corresponding to 5% of TCC).

The values of technical costs used here referred to the official "Listino Tipologico" published by DEI, (Tipografia del Genio Civile) in 2017.

For the Ex Ritiro del Carmine, each rate is shown in Table 6

**Table 6.** Construction cost.

| Construction Cost | € |
|---|---|
| Technical construction cost | 1,445,996.83 |
| Taxes | 303,659.33 |
| Professional costs | 113,727.65 |
| Preliminary studies and surveys | 40,000.00 |
| Tender notice | 5000.00 |
| Accidents | 72,299.84 |
| Total | 1,980,683.66 |

While the cost of construction concerns the early years of the process, the management of the building spans a greater amount of time. Management costs and incomes are evaluated during a time period, which depends on the type and size of the project. For public projects, this is often a period of 20 years [54].

This evaluation requires the formulation of a mode of operation: for each new function, a management hypothesis is made between the "direct management", by the municipality, and the "indirect management", through private rent. Indirect management consists of the renting of the spaces to private companies for each specific activity.

For the monastery, the above hypotheses are reported for each function in Table 7.

**Table 7.** Management model.

| Activity | Management Model |
|---|---|
| Ground floor | |
| Pooled space | Direct |
| Toy room | Direct |
| Association room | Indirect |
| Sacristy | Indirect |
| Restaurant | Indirect |
| Theater | Direct |
| First floor | |
| Antiviolence center | Direct |
| Second floor | |
| Cultural center | Direct |
| Conferences room | Direct |

The costs and the incomes for each function were evaluated through a market analysis and referring to official list prices. The evaluation used both the comparison between historical data, when available, and an indirect approach, when there were no historical data [50].

The evaluated costs were the following:

- Ordinary management (O): linked to necessary things to run the activity;
- Staff management (SM): linked to the costs of the employees;
- Ordinary maintenance (OM): linked to the periodic activities to allow the normal use of the building;
- Extraordinary maintenance (EM): linked to long period for repairs and/or prolonged use of the building;
- Insurance and taxes (IT).

The evaluated incomes were the following:

- Rate income (RI): linked with the tickets sold for public events;
- Nonrate income (NRI): linked with the rent received for the activity management;

The comparison between the costs and the incomes (Table 8) showed that the costs are greater than the incomes. Therefore, it is necessary to find new funds from public capital. These, which are reported in Table 8, derive from participation in public invitations to tender for single activities.

**Table 8.** Costs, incomes, and funds.

|  | Costs | | | | | Incomes | | Funds |
|---|---|---|---|---|---|---|---|---|
|  | **O** | **SM** | **OM** | **EM** | **IT** | **RI** | **NRI** | **F** |
|  | **€/Year** | **k€/Year** | **€/Year** | **€/10Year** | **€/Year** | **€/Year** | **€/Year** | **€/Year** |
| Ground floor | | | | | | | | |
| Pooled spaces | 2400 | 51,662 | 800 | 12,000 | | | | |
| Toy room | 3.6 | 11,310 | 400 | 8000 | | 2400 | | |
| Association room | | | | 5000 | 1155 | | 3000 | |
| Sacristy | | | | 600 | 1680 | | 5400 | |
| Restaurant | | | | 18,000 | 6006 | | 25,200 | |
| Theater | 14,000 | 10,000 | 600 | 16,000 | | 39,230 | 12,500 | 10,000 |
| First floor | | | | | | | | |
| Antiviolence center | 17,850 | 53,192 | 2000 | 50,000 | | | | 91,042 |
| Second floor | | | | | | | | |
| Cultural center | 23,500 | 43,668 | 2500 | 55,000 | | | | 4500 |
| Conferences room | 7000 | | 400 | 8000 | | | 12,000 | |
| **Total** | 68,350 | 169,832 | 6700 | 178,000 | 8841 | 41,630 | 58,100 | 105,542 |

O—ordinary management; SM—staff management; OM—ordinary maintenance; EM—extraordinary maintenance; IT—insurance and taxes; RI—rate income; NRI—nonrate income.

The cash flow is represented on a time line, which relates costs, incomes, and funds to each period of time (Table 9).

The comparison between the costs, incomes, and funds highlights the lack of resources for the management of the Ex Ritiro del Carmine. While the cost of construction is provided by public funds, the incomes do not entirely cover the costs. Therefore, the municipality has to cover the gap with its own resources.

The public financial statement is divided into "missions". The redevelopment of the monastery embraces three different missions, for which €3.3 million are allocated. Therefore, the Ex Ritiro requires at least 1.5% of the stated budget available.

**Table 9.** Cash flow.

| *Values in Thousand-Euros | | | | | | | | | | | | | | | | | | | | |
|---|---|---|---|---|---|---|---|---|---|---|---|---|---|---|---|---|---|---|---|---|
| Time line | 1° | 2° | 3° | 4° | 5° | 6° | 7° | 8° | 9° | 10° | 11° | 12° | 13° | 14° | 15° | 16° | 17° | 18° | 19° | 20° |
| TOTAL COSTS | −1455 | −528 | −313 | −313 | −313 | −314 | −313 | −313 | −313 | −314 | −313 | −424 | −313 | −314 | −313 | −313 | −313 | −314 | −313 | −313 |
| Construction cost | 1455 | 526 | | | | | | | | | | | | | | | | | | |
| Ordinary management | | | 68 | 68 | 68 | 68 | 68 | 68 | 68 | 68 | 68 | 68 | 68 | 68 | 68 | 68 | 68 | 68 | 68 | 68 |
| Staff mangement | | | 170 | 170 | 170 | 170 | 170 | 170 | 170 | 170 | 170 | 170 | 170 | 170 | 170 | 170 | 170 | 170 | 170 | 170 |
| Ordinary maintenance | | | 67 | 67 | 67 | 67 | 67 | 67 | 67 | 67 | 67 | | 67 | 67 | 67 | 67 | 67 | 67 | 67 | 67 |
| Extraordinary maintenance | | | | | | | | | | | | 178 | | | | | | | | |
| Insurance and taxes | | 2 | 7 | 7 | 7 | 9 | 7 | 7 | 7 | 9 | 7 | 7 | 7 | 9 | 7 | 7 | 7 | 9 | 7 | 7 |
| TOTAL INCOMES | 0 | 0 | 100 | 100 | 100 | 100 | 100 | 100 | 100 | 100 | 100 | 100 | 100 | 100 | 100 | 100 | 100 | 100 | 100 | 100 |
| Rate icome | | | 42 | 42 | 42 | 42 | 42 | 42 | 42 | 42 | 42 | 42 | 42 | 42 | 42 | 42 | 42 | 42 | 42 | 42 |
| Non-rate income | | | 58 | 58 | 58 | 58 | 58 | 58 | 58 | 58 | 58 | 58 | 58 | 58 | 58 | 58 | 58 | 58 | 58 | 58 |
| **FLOW (incomes - costs)** | **−1455** | **−528** | **−152** | **−152** | **−152** | **−154** | **−152** | **−152** | **−152** | **−154** | **−152** | **−324** | **−152** | **−154** | **−152** | **−152** | **−152** | **−154** | **−152** | **−152** |
| Public founds | 1455 | 528 | 106 | 106 | 106 | 106 | 106 | 106 | 106 | 106 | 106 | 106 | 106 | 106 | 106 | 106 | 106 | 106 | 106 | 106 |
| **FLOW (incomes - costs + fuonds)** | **0** | **0** | **−47** | **−47** | **−47** | **−48** | **−47** | **−47** | **−47** | **−48** | **−47** | **−218** | **−47** | **−48** | **−47** | **−47** | **−47** | **−48** | **−47** | **−47** |
| Municipality founds | 0 | 0 | 47 | 47 | 47 | 48 | 47 | 47 | 47 | 48 | 47 | 218 | 47 | 48 | 47 | 47 | 47 | 48 | 47 | 47 |
| **Net cash flow** | **0** | **0** | **0** | **0** | **0** | **0** | **0** | **0** | **0** | **0** | **0** | **0** | **0** | **0** | **0** | **0** | **0** | **0** | **0** | **0** |

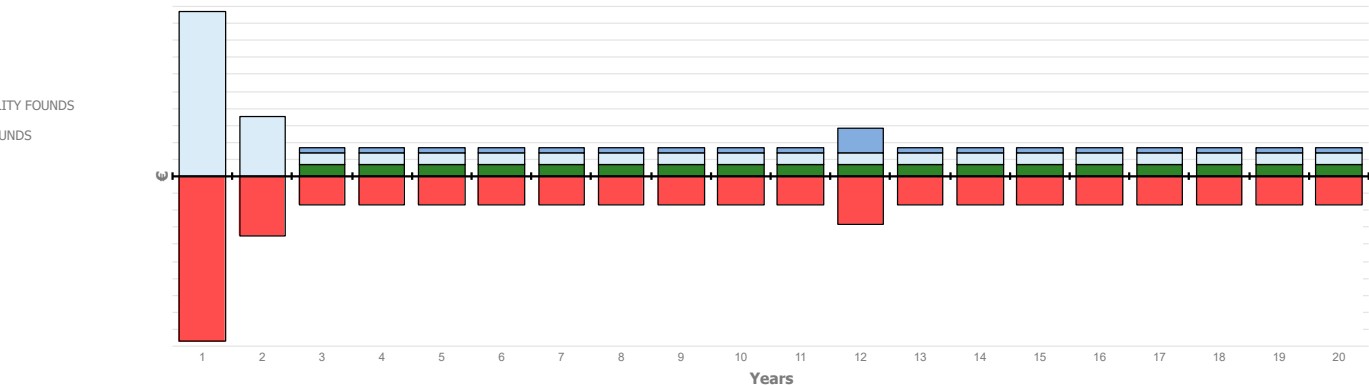

## 4. Discussion and Future Research

This paper presents an integrated methodological approach for the choice of sustainable alternative functions for the adaptive reuse of the former monastery of Ritiro del Carmine in the Campania region from a circular economic perspective.

The proposed methodology integrates different evaluation methods to support the whole decision-making process, from the phase of problem definition to the study of the financial feasibility of the chosen project function, in order to encourage the participation of all stakeholders and to define a "shared solution" that can meet the needs of the local community, as well as be sustainable in the long term from an economic point of view.

The presented case study allowed the exploration of the framework in order to support a flexible and adaptive decision-making process that can consider the complex value of cultural heritage and the multiplicity of actors involved in the decision-making process.

The obtained results showed that "Alternative E: Social hub with antiviolence center for women" was the preferred choice considering the defined evaluation criteria. Moreover, this alternative reached the highest level of consensus among the stakeholders involved in the decision-making process. Nevertheless, the solution found is not sustainable from a financial point of view considering the high investment costs; so, the financial analysis showed the need for access to public funding to cover investment costs. The public investment represents an opportunity to recover a building that is a symbol of the community in order to provide a space for growth and cultural exchange, creating the first cultural center of the city and providing a concrete response to the demand for protection of vulnerable women.

The abovementioned experimentation has shown the usefulness of the integration of different evaluation methods to support the decision-making process in the case of renovation of a heritage building characterized by several elements of complexity. In this sense, the proposed approach seemed appropriate to the case study.

Today, the new challenge for local authorities is to regenerate abandoned heritage buildings while involving different stakeholders and the local community in order to create new models of governance which are able to guarantee economic sustainability and the conservation of historical and cultural values, as in the case of the former monastery of Ritiro del Carmine, which has always been a symbolic building for the community.

From this perspective, future research in this field can be oriented toward the search for new circular financing models, particularly in the field of impact financing, as well as new circular governance models, based on the notion of "heritage as commons", starting from the Italian experimentations of municipal civic agreements for commons management.

**Author Contributions:** Conceptualization: F.T., M.F., and G.A.; Data curation: F.T., M.S. M.F.; Formal analysis: F.T., M.S.; Methodology: F.T.; Validation: F.T., M.F. and G.A.; Writing—original draft: F.T., M.F. G.A. and M.S. All authors read and approved the final manuscript.

**Funding:** This research received no external funding.

**Conflicts of Interest:** The authors declare no conflict of interest.

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
