# Peer review of "An Integrated Decision Support System for the Sustainable Reuse of the Former Monastery of “Ritiro del Carmine” in Campania Region"

_sustainability, doi:10.3390/su11195244_

Round 1

Reviewer 1 Report

General comments

The paper is interesting, in particular by combining different decision-support tools into an integrated approach. However, much has been written about adaptive re-use, including drivers to adaptive re-use and multi-criteria to support decision-making processes. What does this paper add to the existing literature? What research gap or gap in practical support does the paper aims to fill? Why is the proposed method a better one that existing methods (e.g. more valid, reliable, practical applicable, more holistic?). Please explain a bit more.

The methodology should be explained in a much better way, see my remarks below about the methodology.

What lessons have been learned by the case study regarding the decision-making process framework (figure 1)? Is there any reason to adapt the framework in figure 1, based on these lessons learned?

At first sight the paper is well-structured. However, 2.2. Introduction to the case study, includes various sentences (about the necessity of integration of many criteria, stakeholders etc.), that could be more appropriately included in the introduction section.

The paper presents an integrated methodological approach for the choice of sustainable alternative functions for the adaptive re-use of the ex-monastery: I miss a clear explanation of the assessment on sustainability issues such as energy use, CO2 emission etc. It seems that sustainable is defined here as financially sustainable. Please explain.

Please include some pictures of the monastery.

Methodology

Please be more clear about which part of the methodology is based on earlier references and existing methods and which criteria have been added or adapted by the author.

It is not always clear what steps are meant to be generally applicable and what steps are based on particular characteristics of the case that has been studied.

Figure 1, Methodological framework, is very interesting, but should be explained much better, also in connection to related references and in connection to 3.1. Problem definition. Be more precise whether the framework is meant to be a general one or whether it describes what happened in the case study. Besides, does figure 1 present a methodological framework (linked to research) or a decision-making process framework (linked to decision-making) or both: a methodological framework that is being tested in a case study?

Figure 2. Tree charts of criteria, is interesting, but it is not clear to what extent it is based on earlier references and which criteria have been developed by the author.

3.2. Design of the alternative: What “decision-making process has been followed to select these three alternatives? It seems that societal needs were leading here. It also seems that two steps have been applied: first a static vision in which a single function absorbs the whole complex, and second, during the process it arises the possibility of taking advantage of the characteristics of the complex and of a multi-functional building. Please explain the process a bit more clear and explain the connections with the steps in Figure 1 and/or present a flow chart of the decision-making process to select design alternatives.

The impact matrix developed for the case study under analysis is illustrated in Table 1. Here, too, it is not clear which criteria have been adopted from earlier references and existing tools and which criteria have been developed by the author. What does c/b mean in the line sub-criterion – index – c/b?

WHO allocated the weights, based on what arguments? The paper discusses the assessments by a public administrator and a technical expert: why these stakeholders? Are they the main decision makers? Why not other stakeholders, such as the potential investor or new owner?

In the case study under analysis, the ranking has been calculated with the software Definite, Version 3.1.1.6, developed by the Institute for Environmental Studies Vrije Universiteit Amsterdam. HOW does this work?

Figure 3. Sets of ordinal weights: does this figure present the criteria in order of importance?

Figure 4. Results of the Regime method. Please explain a bit more what we see: what do the figures mean? Consider to replace the title of this figure by: Sets of weight based on the Regime assessment.

Table 2. Sets of weight matrix: I don’t understand. For instance: why is cell a x b (1.200) not the same as cell b x a (0.833)? I also don’t understand the step from table 2 to Figure 5. How does it work?

3.3.1 is called Regime method, 3.3.2. is called Wieghted summation (should be weighted summation), which is part of the regime method, and 3.3. is called Social evaluation. Is this part of the Regime method as well, or a new step as part of NAIADE?  Where can I find this step in Figure 1. Methodological framework? Please make better connections between figure 1 and the steps in the text

Ditto 3.3.4. Coalition assessment. Please let the terms in Figure 1 come back in the text and vice versa.

Figure 6 presents a Dendogram, i.e. a graphic representation of the potential coalitions between the parties. Each meeting point has an associated numeric value, named similarity index, which shows the similarity and the credibility of the coalition, with values between 0 and 1. Each index has an associated ranking of the alternatives shared between the members of the coalition. In spite of this explanation I still don’t understand what to see in Figure 6. Hoe are “similarity and credibility of the coalitions” being defined? What do the figures mean? Does a higher figure represent a more “similar” and/or more “credible” coalition?

Please be more consistent in the names of the alternatives, e.g. a or A, b or B etc.

3.3.6. Financial sustainability: please explain a bit more whether/how you could obtain reliable financial data.

Can you give an indication of the time and expertise that is needed to conduct this type of assessment?

Detailed comments

Abstract: it is quite uncommon to included references in an abstract

Avoid literally repetition of sentences in the abstract and the main text.

p. 2 line 65: to favor a circuit of productivity: what do you mean with productivity here?

p. 3 line 135: It is not common to use he/her (or: he/she); usually “he” includes both he and she

p. 4 line 155: a condition of neglection even dereliction: what do you mean to say here?

p. 5 line 196: Starting from the dialog with the DM, the criteria are the technical formulation operated by the analysts and reflects: I don’t understand why it starts with technical formulation, please explain

p. 8 line 325: Figure 3Error! Reference source not found: please check.

p. 10 line 355: Table 2Error! Reference source not found”: please check.

p. 11 line 367: what is a “telematic survey”?

p. 13 line 414: Error! Reference source not found. Please check.

p. 17 line 488: what does “Author Contributions” mean?

References

In the text, refences are referred to by numbers, e.g. [1], [2], etc. whereas the numbers are not included  the list of references at the end of the paper. Please choose either to include the name of the author(s) and the year of the references in the text between parentheses and present the references in alphabetical order, or use numbers in the text to refer to the references at the end and number the references in the list of references accordingly.

The paper is well-documented, but various relevant publications are missing.

Earlier I reviewed related papers for Sustainability, e.g. by Magdalena Celadyn on “Adaptive reuse in environmentally sustainable interior architectural design model” and by Francesca Abastante, Isabella M. Lami and Beatrice Mecca,  on “How to revitalise a historic district: a stakeholders-oriented assessment framework in the adaptive reuse perspective”.  Please check if one of these papers or both have already been published.

Please also check:

Geraedts, R. P., van der Voordt, T. & Remøy, H. (2018), Conversion Potential Assessment Tools. In: Remøy, H. and Wilkinson, S.J. (Eds.) Building Resilience in Urban Settlements through sustainable change of use. Wiley-Blackwell, Chapter 7, 121-151. ISBN 978-1-119-23142-4.

Maria Rita Pinto, Stefania De Medici, Carla Senia, Katia Fabbricatti, and Pasquale De Toro (2017), Building reuse: multi-criteria assessment for compatible design. International Journal of Design Sciences and Technology, Volume 22 Number 2 (2017) ISSN 1630-7267.

Esther H.K. Yung Edwin H.W. Chan , (2015),"Evaluation of the social values and willingness to pay for conserving built heritage in Hong Kong", Facilities, Vol. 33, 1/2 pp. 76 – 98. DOI: http://dx.doi.org/10.1108/F-02-2013-0017

Jane Matthes, Peter E. D. Love (2015), Critical Success Factors of Adapting Heritage Buildings: An Exploratory Study. Built Environment Project and Asset Management · June 2015. DOI: 10.1108/BEPAM-01-2015-0002

Peter A. Bullen and Peter E.D. Love (2011), Adaptive reuse of heritage Buildings. Structural Survey, Vol. 29 No. 5, 2011, pp. 411-421. DOI 10.1108/02630801111182439.

Communication

Please check the text carefully for correct English grammar and spelling, preferably by a native speaker, for instance:

Abstract: although is strongly recognize the effective contribution of cultural heritage as a driver and enabler > rephrase e.g. although the effective contribution of cultural heritage as a driver and enabler .. is strongly recognize, ….

Abstract: investment in cultural heritage have multi dimensional impact > rephrase by either an investment has, or investments have a multi-dimensional impact

Abstract: social, economic and historical and cultural > social, economic, historical and cultural

Abstract: but also stimulate > but also to stimulate Please check the text as well. E.g.

p. 2 line 81: whit the task “with the task;

p. 2 line 86: The capacity of involve several points > either: involving, or: to involve;

p. 2 line 89: the importance of provide systematic information > either importance of providing, or importance to provide;

p. 3 line 105: A survey of the above-mentioned methods > a discussion of the above mentioned methods; etc.

Author Response

Dear reviewere, thank you very much for your comments. I really appreciate them.

Following my explanations to your suggestion/questions. You will find in deep explanations in the text.

The paper is interesting, in particular by combining different decision-support tools into an integrated approach. However, much has been written about adaptive re-use, including drivers to adaptive re-use and multi-criteria to support decision-making processes.

What does this paper add to the existing literature?

The objective of the paper is to propose an integrated methodological framework to support the whole decision making process. The methodology proposed is referred to the decision making theory proposed by Simon 1972, and for each stage identify by Simon, different evaluation methods have been used considering the peculiarity of the case study under analysis.

What research gap or gap in practical support does the paper aims to fill?

We integrate different methods considering the objectives to reach in the case study in each stage of the decision process. In particular we refers also to the financial sustainability considering also the management phase.

Why is the proposed method a better one that existing methods (e.g. more valid, reliable, practical applicable, more holistic?). Please explain a bit more.

The framework proposed tries to follow the decision process in a holistic way from problem definition to decision choice, using different tools in respect to the peculiarity of the case study.

The methodology should be explained in a much better way, see my remarks below about the methodology.

Thank you for your comment I explained better the methodology in paragraph 2.1

What lessons have been learned by the case study regarding the decision-making process framework (figure 1)? Is there any reason to adapt the framework in figure 1, based on these lessons learned?

At first sight the paper is well-structured.

However, 2.2. Introduction to the case study, includes various sentences (about the necessity of integration of many criteria, stakeholders etc.), that could be more appropriately included in the introduction section. OK i changed it in the text

The paper presents an integrated methodological approach for the choice of sustainable alternative functions for the adaptive re-use of the ex-monastery: I miss a clear explanation of the assessment on sustainability issues such as energy use, CO2 emission etc. It seems that sustainable is defined here as financially sustainable. Please explain.

In our case study, the DM showed the intention of pursuing social sustainability-based criteria starting from the literature indication. This vision is elaborated in a holistic view and includes the enhancement of social objectives, economic objectives and urban development [34]. At this strategic stage of the project development no specific environmental issues have been considered, but these will be included later in the design project phase. The hierarchical structure of the criteria was formulated during a focus-group in which the different points of view arose and became clearer: the board session allowed the analyst to summarize the information in a limited number of clearly-defined criteria.

Please include some pictures of the monastery.

Ok I have added two pictures

Methodology

Please be more clear about which part of the methodology is based on earlier references and existing methods and which criteria have been added or adapted by the author.

We referes to the litterature for the definition of main criteria (social, economic and urban development) than the hierarchical structure of the criteria was formulated during a focus-group in which the different points of view arose and became clearer: the board session allowed the analyst to summarize the information in a limited number of clearly-defined criteria.

It is not always clear what steps are meant to be generally applicable and what steps are based on particular characteristics of the case that has been studied.

Following the classical approach proposed by Simon (1972) to the decision-making process, an integrated methodological framework is proposed for the case study under analysis as figure 1 illustrates.

Simon defines three main stages of the decision-making process:

Intelligence, which deals with the problem of identification and the data collection of the problem. Design, which deals with the generation of alternative solutions to the problem at hand. Choice, that is selecting the “best” solution from among the alternative solutions using some criterion. 

Starting from the main stages proposed by Simon, different evaluation methods have been proposed in order to support the definition of different re-use alternatives in the redevelopment projects, including both technical problems related to the design projects [22], social issues and the financial feasibility. In the stated methodological framework different multicriteria methods are combined with a financial model in the diverse stages of the decision-making process in order to develop a tool to enhance the quality and the trustworthiness of the decision-making process itself.

Figure 1, Methodological framework, is very interesting, but should be explained much better, also in connection to related references and in connection to

The methodology follows the general approach to decision problem [Simon 1972], adapted to the case study analyzed in order to support the public administration - the owner of the monastery - to choose the best alternative re-use functions in a sustainable perspective. (line 86-88 and paragraph 2.1)

3.1. Problem definition. Be more precise whether the framework is meant to be a general one or whether it describes what happened in the case study. Besides, does figure 1 present a methodological framework (linked to research) or a decision-making process framework (linked to decision-making) or both: a methodological framework that is being tested in a case study?

Following the classical approach proposed by Simon (1972) to the decision-making process, an integrated methodological framework is proposed for the case study under analysis as figure 1 illustrates.

Simon defines three main stages of the decision-making process:

Intelligence, which deals with the problem of identification and the data collection of the problem. Design, which deals with the generation of alternative solutions to the problem at hand. Choice, that is selecting the “best” solution from among the alternative solutions using some criterion. 

Starting from the main stages proposed by Simon, different evaluation methods have been proposed in order to support the definition of different re-use alternatives in the redevelopment projects, including both technical problems related to the design projects [22], social issues and the financial feasibility. In the stated methodological framework different multicriteria methods are combined with a financial model in the diverse stages of the decision-making process in order to develop a tool to enhance the quality and the trustworthiness of the decision-making process itself.

Figure 2. Tree charts of criteria, is interesting, but it is not clear to what extent it is based on earlier references and which criteria have been developed by the author.

The hierarchical structure of the criteria is formulated during a focus-group in which the different points of view arise and became clearer: the board session allows the analyst to summarize the several information in a limited number of clearly defined criteria. For the analysed case study, the formulated criteria are represented in a “tree chart”, as showed in Figure 2.

3.2. Design of the alternative: What “decision-making process has been followed to select these three alternatives? It seems that societal needs were leading here. It also seems that two steps have been applied: first a static vision in which a single function absorbs the whole complex, and second, during the process it arises the possibility of taking advantage of the characteristics of the complex and of a multi-functional building. Please explain the process a bit more clear and explain the connections with the steps in Figure 1 and/or present a flow chart of the decision-making process to select design alternatives.

In the early stage of the evaluation process, with respect to the objectives of the public administration, designer formulated three alternatives.  

These three alternatives were presented in a focus group to the public administration and the new idea emerged in a brainstorming section considering the characteristic of the building and the objectives of the public administration.

Actually, during the deliberation process the possibility of taking advantage of the characteristics of the complex and of a multi-functional building was suggested: the integration of different functions and resources in a hybrid building is able to enhance the social services provided and to reduce public expenditure.

Two more alternative were proposed.

The five developed alternatives were then evaluated in a second stage of the decision process. They are evaluated from a technical point of view considering the list of criteria developed and with the help of Regime analysis and Weighted Summation method. Then the social acceptability of the solution found was evaluated with the NAIADE methods. The results obtained are reported in the next paragraph.

The impact matrix developed for the case study under analysis is illustrated in Table 1. Here, too, it is not clear which criteria have been adopted from earlier references and existing tools and which criteria have been developed by the author. What does c/b mean in the line sub-criterion – index – c/b?

We referes to the litterature for the definition of main criteria (social, economic and urban development) than the hierarchical structure of the criteria was formulated during a focus-group in which the different points of view arose and became clearer: the board session allowed the analyst to summarize the information in a limited number of clearly-defined criteria.

For each of the criteria, it is indicated whether the criterion represents a cost or a benefit (c/b).

WHO allocated the weights, based on what arguments? The paper discusses the assessments by a public administrator and a technical expert: why these stakeholders? Are they the main decision makers? Why not other stakeholders, such as the potential investor or new owner?

As part of the Public Estate, the DM is represented by the Municipality itself whit the position of main investor.

(The municipality is the owner and the investor, the other stakeholder are considered in the next steps to give the DM useful and more precise information to gain better choice.)

The main stakeholder has been identify during a focus group with the public administration considering the strtaegic objective of the municiplity

In the case study under analysis, the ranking has been calculated with the software Definite, Version 3.1.1.6, developed by the Institute for Environmental Studies Vrije Universiteit Amsterdam. HOW does this work?

The Definite programme was developed by Ron Janssen and Marine van Herwijnen in 1987 [50]. It’s a multiojective decision support system that supports the whole decision process from problem definition to report generation. The system performs the following functions: 1) structure the problem and generate alternative; 2) compare alternative; 3) rank and or value alternative; 4) support interpretation of the results; 5) Presents results. To perform these functions the system contains five module. Each module contains a variety of procedures (like Regime and Weight Summation Methods) to perform these functions. (line 177-184 pag. 5)

Figure 3. Sets of ordinal weights: does this figure present the criteria in order of importance? Yes I change the title of the figure

Figure 4. Results of the Regime method. Please explain a bit more what we see: what do the figures mean? Consider to replace the title of this figure by: Sets of weight based on the Regime assessment.

The figure show the rank order of the alternative. I change the title of the figure

Table 2. Sets of weight matrix: I don’t understand. For instance: why is cell a x b (1.200) not the same as cell b x a (0.833)? I also don’t understand the step from table 2 to Figure 5. How does it work?

The matrix is not symmetrical, because if we compare a with b than the value of b in respect to a represent the reciprocal. For example, in our case the criterion a is more important than b and the value is 1.2. So the value of b in respect to a is the reciprocal (1/1.2) 0,833.

3.3.1 is called Regime method, 3.3.2. is called Wieghted summation (should be weighted summation), yes

which is part of the regime method, and 3.3. is called Social evaluation. Is this part of the Regime method as well, or a new step as part of NAIADE?  Where can I find this step in Figure 1. Methodological framework? Please make better connections between figure 1 and the steps in the text

Regime method is a multiciteria analysis used in order to evaluate the rank order of the alternative. NAIADE in also a multicriteria analysis but is not part of the Regime. We use the NAIADE as social analysis to understand the level of coalition between the stakeholders on the choice of the alternative. I change the figure and I hope that now is more clear.

I explain better the methodology in the text (see par 2.1)

Ditto 3.3.4. Coalition assessment. Please let the terms in Figure 1 come back in the text and vice versa. OK

Figure 6 presents a Dendogram, i.e. a graphic representation of the potential coalitions between the parties. Each meeting point has an associated numeric value, named similarity index, which shows the similarity and the credibility of the coalition, with values between 0 and 1. Each index has an associated ranking of the alternatives shared between the members of the coalition. In spite of this explanation I still don’t understand what to see in Figure 6. Hoe are “similarity and credibility of the coalitions” being defined?

The first coalition, between G1-G6 (public administration and social and cultural association), takes place for a value of the similarity index of 0.8545, which represent an 85% of similarity between the group’s preferences. The second coalition, between G4-G7 (freelance professional and G7 student), takes place for a value of the similarity index of 0.8171. The third coalition, between G8 (employed) and the previous coalition (G4-G7), takes place for a value of 0.8058. The coalition between the two above-mentioned groups G1-G6 and G4-G7-G8 takes place for a value of 0.7954.

What do the figures mean?

Does a higher figure represent a more “similar” and/or more “credible” coalition?

Represent the level of similarity between group preferences. As explained at the previous point each level represent a coalition between different groups referred to the rank order of the alternative

Please be more consistent in the names of the alternatives, e.g. a or A, b or B etc. ok

3.3.6. Financial sustainability: please explain a bit more whether/how you could obtain reliable financial data.

There are three specific types based on the degree of definition of a process industry plant: 1) Order of Magnitude Estimate, 2) Budget Estimate and 3) Definitive Estimate.

Typically an estimate is an assessment, based on specific facts and assumptions, of the final cost of a project, program, product, or process.
The results of this process vary with the:

type, amount, and accuracy of scope and estimating data available phase of the project  time allotted to prepare the estimate 
  perspective of valuation (contractor, designer, owner) 
  skill of the estimator 
  calculation technique 
  cost accuracy desired 

In the case under analysis that is a feasibility study stage as the literature suggests we refers to an approximate estimate of costs and revenues referring to ordinary values published by official list prices (eg Typological list price for costs published by DEI every years and Data from Italian Territorial Agency for revenues)

Can you give an indication of the time and expertise that is needed to conduct this type of assessment?

The expertise required can be oriented to economics skills but integrated to designer skills more oriented to the knowledge of the project. The time depends on the complexity of the case study and the availability of data.

Detailed comments

Abstract: it is quite uncommon to included references in an abstract.

Thank you for your comment I included the references in the main text and not in the abstract.

Avoid literally repetition of sentences in the abstract and the main text. OK

2 line 65: to favor a circuit of productivity: what do you mean with productivity here?

I change the line with: Therefore, a sustainable project for the re-use of an historical building is a multidimensional issue which has to respect its specific identity and historical values but at the some time the choice of function should guarantee its management during the time.

3 line 135: It is not common to use he/her (or: he/she); usually “he” includes both he and she

Thank you I use he

4 line 155: a condition of neglection even dereliction: what do you mean to say here?

I change the sentence with: “In the last years the difficulties to find new uses for these properties lead to a condition of abandonment and ruin”.

5 line 196: Starting from the dialog with the DM, the criteria are the technical formulation operated by the analysts and reflects: I don’t understand why it starts with technical formulation, please explain

8 line 325: Figure 3Error! Reference source not found: please check. ok 10 line 355: Table 2Error! Reference source not found”: please check. ok 11 line 367: what is a “telematic survey”? on line survay 13 line 414: Error! Reference source not found. Please check. ok 17 line 488: what does “Author Contributions” mean?

This is required by the journal to define the single contributions of the author to the paper. I think that this is shown in the final version of the paper.

References

In the text, refences are referred to by numbers, e.g. [1], [2], etc. whereas the numbers are not included  the list of references at the end of the paper. Please choose either to include the name of the author(s) and the year of the references in the text between parentheses and present the references in alphabetical order, or use numbers in the text to refer to the references at the end and number the references in the list of references accordingly.

I add the number at the end in the reference list

The paper is well-documented, but various relevant publications are missing.

Thank you very much for your suggestion. I add the following refrences

Earlier I reviewed related papers for Sustainability, e.g. by Magdalena Celadyn on “Adaptive reuse in environmentally sustainable interior architectural design model” and by Francesca Abastante, Isabella M. Lami and Beatrice Mecca,  on “How to revitalise a historic district: a stakeholders-oriented assessment framework in the adaptive reuse perspective”.  Please check if one of these papers or both have already been published.

Please also check:

Geraedts, R. P., van der Voordt, T. & Remøy, H. (2018), Conversion Potential Assessment Tools. In: Remøy, H. and Wilkinson, S.J. (Eds.) Building Resilience in Urban Settlements through sustainable change of use. Wiley-Blackwell, Chapter 7, 121-151. ISBN 978-1-119-23142-4.

Maria Rita Pinto, Stefania De Medici, Carla Senia, Katia Fabbricatti, and Pasquale De Toro (2017), Building reuse: multi-criteria assessment for compatible design. International Journal of Design Sciences and Technology, Volume 22 Number 2 (2017) ISSN 1630-7267. Metodologia

Esther H.K. Yung Edwin H.W. Chan , (2015),"Evaluation of the social values and willingness to pay for conserving built heritage in Hong Kong", Facilities, Vol. 33, 1/2 pp. 76 – 98. DOI: http://dx.doi.org/10.1108/F-02-2013-0017

Jane Matthes, Peter E. D. Love (2015), Critical Success Factors of Adapting Heritage Buildings: An Exploratory Study. Built Environment Project and Asset Management · June 2015. DOI: 10.1108/BEPAM-01-2015-0002 (usare per giustificare scelta dei criteri)

Peter A. Bullen and Peter E.D. Love (2011), Adaptive reuse of heritage Buildings. Structural Survey, Vol. 29 No. 5, 2011, pp. 411-421. DOI 10.1108/02630801111182439.

Communication OK

Please check the text carefully for correct English grammar and spelling, preferably by a native speaker, for instance:

Thank you for your suggestion. The paper has been carefully correct by a native speaker.

Abstract: although is strongly recognize the effective contribution of cultural heritage as a driver and enabler > rephrase e.g. although the effective contribution of cultural heritage as a driver and enabler .. is strongly recognize, …. OK

Abstract: investment in cultural heritage have multi dimensional impact > rephrase by either an investment has, or investments have a multi-dimensional impact OK

Abstract: social, economic and historical and cultural > social, economic, historical and cultural OK

Abstract: but also stimulate > but also to stimulate Please check the text as well. E.g. OK

2 line 81: whit the task “with the task; ok 2 line 86: The capacity of involve several points > either: involving, or: to involve; ok 2 line 89: the importance of provide systematic information > either importance of providing, or importance to provide; ok 3 line 105: A survey of the above-mentioned methods > a discussion of the above mentioned methods; etc. ok

Please check line 414 – “For the Ex Ritiro del Carmine each rate is showed in Error! Reference source not found..

Line 375 – ERROR! Where one can read “The collected evaluations are gathered in the equity matrix showed in Table 4”, one should read “The collected evaluations are gathered in the equity matrix showed in Table 3.”

Lines 420-422 – Authors wrote: “This evaluation requires the formulation of a mode of operation: for each new function it’s made a management hypothesis between the “direct management”, by the municipality, and the “indirect management”, through private rent.” For a better understanding of this idea, how does the indirect management mode relates with the private rent? It must be explained

Reviewer 2 Report

Dear Authors,

The article focuses on an updated and interesting theme - the discussion around the circular economy and sustainability of the built and cultural heritage - where much remains to be investigated.

COMMENTS

Abstract

The abstract is clear. Nevertheless, the main objective of the manuscript should be present (preferably in the beginning of the Abstract section) which hasn´t been done. Only in lines 26-29, when mentioning the results, authors refer for the first time to an "objective", creating doubt in the reader if this mentioned objective is just the objective of the model or, by the contrary, if it is the main objective of the article.

Please note that only in the Introduction section (lines 67-72) the Main objective is clear identified:

The paper proposes an integrated evaluation model, based on multicriteria analysis, to support the choice of the alternative re-use function of an ancient monastery in the municipality of Mugnano in the Campania Region, (…)”.

Keywords

The presented keywords are: 1 Integrated evaluation; 2  Multicriteria analysis; 3. Cultural heritage and circular economy; 4. Financial sustainability

Please note the punctuation error: “1 Integrated evaluation 2; Multicriteria analysis; (…)”.

When submitting a manuscript, keywords should be well selected and well-focused, considering the central topic of the manuscript and its case study. In this sense, keywords should be efficient and bring precise information about what the manuscript is related with.

Suggestion – if possible, please add the following composed keywords: Ritiro del Carmine; built heritage sustainable re-use.

Composed keywords reveal to be very useful for readers get familiar with the topic of the manuscript.

Introduction

Even considering the mention of 6 literature references, the Introduction would benefit much from a larger number of references in this section, in particular.

The integrated decision support System for the choice of alternative functions (lines 78-151)

Is there any particular reason to write the word System with capital letter?

Introduction to the case study (lines 152-192)

Given the historical importance of the monastery, and mostly being it the case study, wouldn't it deserve at least one photo, plans, sketches… in this section?

Lines 206-219. Based on reference [40] which is not well identified in the References, authors listed 11 representative groups that were selected to address all the social components involved in the project. However, it is not clear where “public services” enter. In this list, one can’t forget public services as it represent an important role in cultural heritage sustainability. How Authors do consider this? OR, do Authors assume that “public services” are imbibed in the selected group “public administration”?

Wouldn’t also tourists or visitors eventually a group to be considered in the list? Especially if the area is very popular with tourists/visitors…

Figure 2 - Please note in the title of 2nd column the presented word SUB_CRITERIa

Lines 224-225 – After Figure 2, when referring the evaluation sub-criteria, shouldn’t the word “organization” come in plural?

Line 313 - Yes/no. binary scale – incorrect spelling punctuation.

Please check line 325 – “(…) are showed below in Figure 3Error! Reference source not found”..

Please check line 355 – “(…) The identified weight matrixes are showed below in Table 2Error! Reference source not found”.

Please check line 414 – “For the Ex Ritiro del Carmine each rate is showed in Error! Reference source not found..

Line 375 – ERROR! Where one can read “The collected evaluations are gathered in the equity matrix showed in Table 4”, one should read “The collected evaluations are gathered in the equity matrix showed in Table 3.”

Lines 420-422 – Authors wrote: “This evaluation requires the formulation of a mode of operation: for each new function it’s made a management hypothesis between the “direct management”, by the municipality, and the “indirect management”, through private rent.” For a better understanding of this idea, how does the indirect management mode relates with the private rent? It must be explained.

Author Response

Dear reviewere, thank you very much for your comments. I really appreciate them.

Following my explanations to your suggestion/questions. You will find in deep explanations in the text.

Dear Authors,

The article focuses on an updated and interesting theme - the discussion around the circular economy and sustainability of the built and cultural heritage - where much remains to be investigated.

COMMENTS

Abstract

The abstract is clear. Nevertheless, the main objective of the manuscript should be present (preferably in the beginning of the Abstract section) which hasn´t been done. Only in lines 26-29, when mentioning the results, authors refer for the first time to an "objective", creating doubt in the reader if this mentioned objective is just the objective of the model or, by the contrary, if it is the main objective of the article.

Thank you for your comment. I underline the objective of the paper in the abstract

Please note that only in the Introduction section (lines 67-72) the Main objective is clear identified:

The paper proposes an integrated evaluation model, based on multicriteria analysis, to support the choice of the alternative re-use function of an ancient monastery in the municipality of Mugnano in the Campania Region, (…)”.

Keywords

The presented keywords are: 1 Integrated evaluation; 2  Multicriteria analysis; 3. Cultural heritage and circular economy; 4. Financial sustainability

Please note the punctuation error: “1 Integrated evaluation 2; Multicriteria analysis; (…)”.

Thank you for your comments I changed the punctuation

When submitting a manuscript, keywords should be well selected and well-focused, considering the central topic of the manuscript and its case study. In this sense, keywords should be efficient and bring precise information about what the manuscript is related with.

Suggestion – if possible, please add the following composed keywords: Ritiro del Carmine; built heritage sustainable re-use.

Thank you for your suggestion I added the composed keywords

Composed keywords reveal to be very useful for readers get familiar with the topic of the manuscript.

Introduction

Even considering the mention of 6 literature references, the Introduction would benefit much from a larger number of references in this section, in particular.

The integrated decision support System for the choice of alternative functions (lines 78-151)

Thank you for your suggestion I add the following literature references:

Geraedts, R. P., van der Voordt, T. & Remøy, H. (2018), Conversion Potential Assessment Tools. In: Remøy, H. and Wilkinson, S.J. (Eds.) Building Resilience in Urban Settlements through sustainable change of use. Wiley-Blackwell, Chapter 7, 121-151. ISBN 978-1-119-23142-4.

Maria Rita Pinto, Stefania De Medici, Carla Senia, Katia Fabbricatti, and Pasquale De Toro (2017), Building reuse: multi-criteria assessment for compatible design. International Journal of Design Sciences and Technology, Volume 22 Number 2 (2017) ISSN 1630-7267. Metodologia

Esther H.K. Yung Edwin H.W. Chan , (2015),"Evaluation of the social values and willingness to pay for conserving built heritage in Hong Kong", Facilities, Vol. 33, 1/2 pp. 76 – 98. DOI: http://dx.doi.org/10.1108/F-02-2013-0017

Jane Matthes, Peter E. D. Love (2015), Critical Success Factors of Adapting Heritage Buildings: An Exploratory Study. Built Environment Project and Asset Management · June 2015. DOI: 10.1108/BEPAM-01-2015-0002 (usare per giustificare scelta dei criteri)

Peter A. Bullen and Peter E.D. Love (2011), Adaptive reuse of heritage Buildings. Structural Survey, Vol. 29 No. 5, 2011, pp. 411-421. DOI 10.1108/02630801111182439.

Is there any particular reason to write the word System with capital letter? No I change it

Introduction to the case study (lines 152-192)

Given the historical importance of the monastery, and mostly being it the case study, wouldn't it deserve at least one photo, plans, sketches… in this section? Thank you I add two photos

Lines 206-219. Based on reference [40] which is not well identified in the References, authors listed 11 representative groups that were selected to address all the social components involved in the project. However, it is not clear where “public services” enter. In this list, one can’t forget public services as it represent an important role in cultural heritage sustainability. How Authors do consider this? OR, do Authors assume that “public services” are imbibed in the selected group “public administration”? yes is included

Wouldn’t also tourists or visitors eventually a group to be considered in the list? Especially if the area is very popular with tourists/visitors… we do not include tourist or visitors because the municipality Mugnano di Napoli hasn’t a touristic nature. The main objective to reach with requalification of the historical building is social and referred to local community.

Figure 2 - Please note in the title of 2nd column the presented word SUB_CRITERIa OK changed

Lines 224-225 – After Figure 2, when referring the evaluation sub-criteria, shouldn’t the word “organization” come in plural? Yes thank you

Line 313 - Yes/no. binary scale – incorrect spelling punctuation. Ok thank you i changed in the text

Please check line 325 – “(…) are showed below in Figure 3Error! Reference source not found”.. ok

Please check line 355 – “(…) The identified weight matrixes are showed below in Table 2Error! Reference source not found”. Ok

Round 2

Reviewer 1 Report

General comments

The paper has been improved by adopting many of my comments. However, I still have various comments that should be coped with. In particular the (lack of) consistency between the steps in Figure 1 and the steps in the text needs further consideration.

Take care that Figure 1. Methodological framework, is well readable. Besides, section 2.1. The integrated decision support system for the choice of alternative functions, describes briefly all analyses conducted in the case study, which are described in more detail in paragraph 3. Please show more clearly how all analyses fit into Figure 1. For instance, the text mentions two analyses: technical and social, whereas Figure 1 mentions also a financial feasibility analysis. Where can the on line survey supplied through Google Forms (line 454) be found in Figure 1? Where can the dendogram (line 469) be found in Figure 1? Section 3.3.5. Financial sustainability, is presented as part of Section 3, Multi criteria assessment, whereas Figure 1 presents the financial feasibility assessment as part of phase 4, Definition of a project strategy. Consider to rename 3.3.5, Financial feasibility, by 3.4 Definition of a project strategy, or rename the box in Figure 1.

Besides, various statements and tables needs further clarification.

Line 104: different evaluation methods have been proposed > include references, [22] is insufficient. For instance [21] is appropriate here as well

Line 126: The alternative solution was evaluated by means of a multicriteria analysis considering two type of analysis: i) a technical analysis > two types of analysis: what is type ii? Only in line 181 we see ii) social analysis. This is confusing. Line 142 states: This conception implies the necessity of two types of evaluation [31, 32]: a “technical evaluation” linked with the DM’s preferences and a “social evaluation” linked with the stakeholders’ points of view. > please check the text on consistency and clarity and avoid repetitions.

Line 147: The Regime method > please add a reference

Line 166: impact matrix: please define

Line 242: starting from the literature indication > please rephrase, e.g. by mentioning the most appropriate references.

Line 244: At this strategic stage of the project development no specific environmental issues have been considered, but these will be included later in the design project phase. Why not in the problem definition phase?

Line 246: during a focus-group > rephrase: by a focus group, or: during a focus group session. What literature has been used in this phase and what criteria have been used to define the formulated criteria that are represented in a “tree chart”, as shown in Figure 3.?

Line 284: The representative groups are selected to address all the social components involved in the project > selected by whom?

Line 301: designer formulated three alternatives > the designer, or: the designers? What was the role of all stakeholders in this selection?

Line 435: I still don’t understand the meaning of a, b, c … etc. in Table 2, Sets of weight matrix.

Line 455: The questionnaire is structured with close-ended questions; they have a limited number of answers among which the respondent must choose the one which best matches his/her opinion. > please provide an example to give the reader an idea of the type of questions

Line 484: The dendogram is difficult to understand for readers who are not familiar with it. I still don’t understand what to see in Figure 6. Hoe are “similarity and credibility of the coalitions” being defined? What do the figures mean? Does a higher figure represent a more “similar” and/or more “credible” coalition?

Line 488: Table 4. Ranking of the coalitions, is also difficult to understand.

Line 510 presents Table 6. Construction cost. (or: costs?) > are these the original construction costs when this monastery was built? Or the costs when the monastery would be built now? Or the construction costs of the best compromise solution “Alternative E: Social Hub with anti-violence center for women”? Same question regards Table 9. Cash Flow.

Please add a legend to Table 8; O =, SM =, OM = etc. How reliable are the figures in this Table? Based on what information?

Line 556: Ex Ritiro requires at least 1.5% of the stated available > I don’t understand

What lessons have been learned by the case study regarding the decision-making process framework (figure 1)? Is there any reason to adapt the framework in figure 1, based on these lessons learned?

The paper presents an integrated methodological approach for the choice of sustainable alternative functions for the adaptive re-use of the ex-monastery: I miss a clear explanation of the assessment on sustainability issues such as energy use, CO2 emission etc. Please explain a bit more.

Can you give an indication of the time and expertise that is needed to conduct this type of assessment?

Please have a final check on correct English grammar and sentences. A few examples:

Line 48: economic and historical and cultural > economic, historical and cultural

Line 49: can contribute an increase in overall local productivity > can contribute to …

Line 50: and attract funds > and attracting funds

Line 72: The choice of the new function to be made > skip “to be made”

Line 73: give the information to identify the best solution > please skip “give” and rephrase into “sufficient information in order to be able to identify the best solution”

Line 77: the objective of the paper is to proposes > to propose

Line 83: general approach to decision problem > decision problems

Line 118: the literature indication [26] > replace by: reference [26]

Line 281: in to > into

Line 325: The above present alternatives > above presented alternatives

Line 326: These three alternatives were presented in a focus group to the public administration: a focus group that included the public administration? Or: in a focus group and after discussion also to the public administration? It is not easy to understand what happened precisely.

Author Response

Dear reviewer, here you find attach my reply to you comments.

I reply all the point and I hope that now all is clear.

Thank you very much for all the comments that improve the quality of my paper
